| Editor's Pick | Public Health | Perspective

# The case for microbiome stewardship: what it is and how to get there

Mallory J. Choudoir,[1] Suzanne L. Ishaq,[2] Robert G. Beiko,[3] Diego S. Silva,[4] Emma Allen-Vercoe,[5] Kieran C. O'Doherty[6]

**ABSTRACT** Microbiomes are essential for human, animal, plant, and ecosystem health. Despite widespread recognition of the importance of microbiomes, there is little attention paid to monitoring and safeguarding microbial ecologies on policy levels. We observe that microbiomes are deteriorating owing to practices at societal levels such as pesticide use in agriculture, air and water pollution, and overuse of antibiotics. Potential policy on these issues would cross multiple domains such as public health, environmental protection, and agriculture. We propose microbiome stewardship as a foundational concept that can act across policy domains to facilitate healthy microbiomes for human and ecosystem health. We examine challenges to be addressed and steps to take toward developing meaningful microbiome stewardship.

**KEYWORDS** human microbiome, microbiome stewardship, microbial ecology, ethics, conservation, public health, policy, ecosystem health

The central importance of microbiomes and their ecological functions (e.g., global nutrient cycling, decomposition, bioremediation, pathogen reduction, and as a source of food/nutrition and critical metabolites/molecules) for the health and well-being of ecosystems, animal, plant, and human health is now well established. And yet, alarming evidence is accumulating of adverse health outcomes associated with microbiome deterioration, and for humans, this includes conditions such as asthma, inflammatory bowel disease, and cystic fibrosis. In these clinical contexts, there is an appropriate focus on strengthening an individual patient's microbiome through pharmaceutical, dietary, or other interventions for improved health outcomes.

However, little attention has been given to the broad factors that cause the deterioration of microbiomes in the first place. The lack of meaningful and holistic policy efforts to assess microbial ecology and to monitor potentially detrimental practices is, therefore, disconcerting. Factors adversely affecting microbiomes cross multiple regulatory domains including public health, agriculture and environmental protection. We therefore need a unifying concept emphasizing the importance of microbiomes for human and environmental health that can be applied across policy domains.

Here, we provide an overview of evidence suggesting that there is a deterioration of microbiomes across ecosystems. We argue that these trends need to be appreciated and understood at the societal level (e.g., food systems, sanitation, and built environment), which implies that interventions for sustained microbiome health must also be made at the policy level as opposed to the individual level of treatment and/or behavioral decisions. We provide examples of specific areas where policy interventions should be considered, and we discuss the concept of microbiome stewardship as a foundation for the development of future policies.

**Peer Reviewer** Lita Proctor, Lita Proctor, National Institutes of Health, Bethesda, Maryland, USA

Address correspondence to Kieran C. O'Doherty, odohertk@uoguelph.ca.

The authors declare no conflict of interest.

See the funding table on p. 6.

## SOCIETAL FACTORS THAT LEAD TO MICROBIOME DETERIORATION

Illnesses related to ecological shifts in microbiomes are increasing globally, hinting at an overall deterioration of our shared microbial environments from which our microbiomes are sourced. Many factors beyond the level of individual choice and behavior shape human microbiomes (Fig. 1), such as interactions in shared spaces (1), cultural and dietary practices, food systems and industrialized food processes (2, 3), quality of natural (4) and built environments, and sanitation or pollution (5). For instance, exposure to more natural ecosystems, including urban areas with sufficient biodiverse greenspaces, provides critical microbiota and supports our immune system (6). Conversely, polluted environments can amplify pathogen exposures or create ecosystem shifts that cause disease (7, 8). Exposures during infancy and early childhood are especially critical for shaping human microbiomes (8). These adverse health outcomes also associate with and amplify socioeconomic and racial disparities (9).

Diet is an important influence on gut microbiomes and their related effects on health, and current efforts to improve gut health typically focus on personal dietary interventions. However, the ability to choose one's diet is highly dependent on socioeconomic and geographical factors as well as regional food systems, including variations in agricultural production, food processing, and food access (2, 3). Interventions aimed to improve gut microbiome health at the population level need to operate at broader food system levels and need to consider the impacts of food system practices—from production to processing to intake—on microbial ecology.

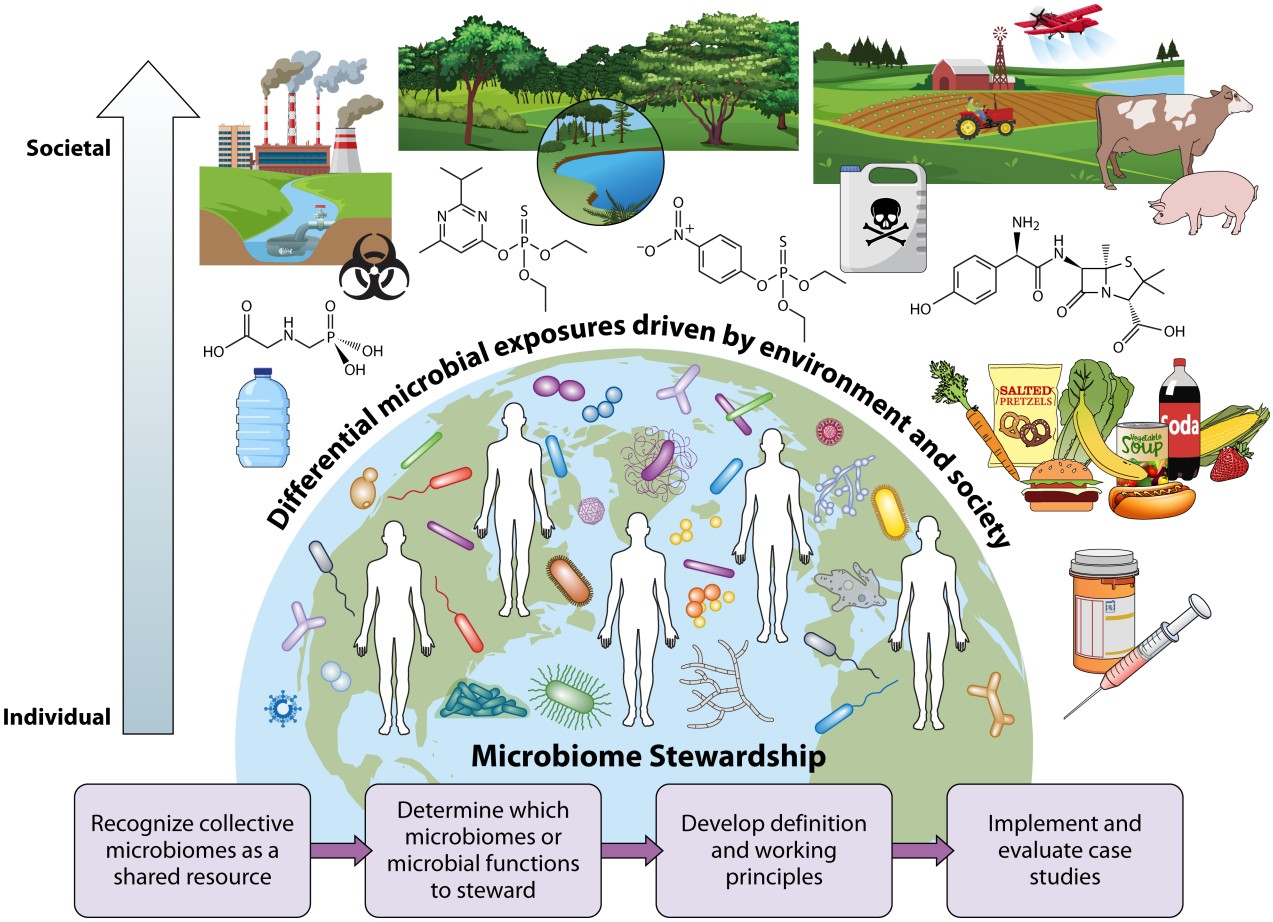

**FIG 1** Microbiome stewardship as a concept and framework for ensuring human and planetary health supported by microbial functions. Human microbiomes are constituted from our environment, which has determinants based largely on societal systems (e.g., agriculture and food systems, built environment, and health-care accessibility) that operate beyond individual choice and behavioral interventions.

In addition to material determinants, social factors also affect the constitution of human microbiomes. For example, racial categorizations are context-specific social constructs by historical era and geographic location and are not biological drivers of the microbiome. Yet, the human microbiome is often presented as being shaped by fallible definitions of ancestry and skin pigmentation. Thus, race is a "ghost variable" (10), which is used as a convenient, but wildly imprecise, proxy for lived experience. Nevertheless, systematic and institutional discrimation of social groups based on skin color and ancestry (i.e., racism) does lead to adverse chemical, microbial, and stress exposures (9), as well as a lack of high-quality food and resources in underserved communities. Therefore, although "race" is not a meaningful biological category for humans, historical and perpetuated racialization/racism and its consequences can have dramatic effects on microbiomes related to adverse health outcomes (10).

## THE POLICY GAP

The deep-rooted interconnections between microbes, humans, and environments suggest that effective interventions for improved health must consider the collective microbiome, which necessitates consideration of a broader ecological and societal context (4, 11, 12). Although the importance of microbes has recently been recognized by several high-level agreements and initiatives (e.g., United Nations Convention of Biological Diversity, Food and Agricultural Organization International Network on Soil Biodiversity, Intergovernmental Panel on Climate Change, One Health approaches), we lack a comprehensive understanding of the explicit roles microbes occupy across ecological and temporal scales. At the moment, there are no meaningful policy initiatives that focus on the importance of microbial ecologies for human, animal, plant, or ecosystem health, although opportunities exist.

In 2022, a United Nations General Assembly resolution declared a right to "healthy environment, including clean air, water, and stable climate." We have some clear criteria for removing environmental contamination and mitigating antimicrobial resistance in soil and wastewater, but there are abundant opportunities to protect microbial ecologies that support healthy environments. For example, many jurisdictions have laws mandating clean air, but a microbial lens would add extra weight to environmental protections given that air pollution is associated with the alteration of the gut microbiome (5).

In food and nutrition policy, attention is focused on macronutrients, yet evidence suggests that food production practices shape both the microbiomes of foods and human microbiomes with no awareness of these dynamics in food or agricultural policies. Growing evidence shows that chronic exposure to agricultural pesticides (i.e., insecticides, herbicides, and fungicides) leads to gut microbiome disturbances (13). Food additives, such as emulsifiers, cause changes in gut microbiomes that are linked to inflammation and metabolic syndrome (14). Conversely, for example, traditionally (Indigenous Greenlandic) prepared foods have much richer microbiome profiles than the same foods prepared with industrial methods (2).

## CONCEPT OF MICROBIOME STEWARDSHIP

We are not lone champions of microbial protection and conservation. "One Health" principles recognize the importance of microbes in human, animal, and plant health. Microbiologists and climate change experts have issued a consensus statement calling attention to the role of microbes in climate change responses (15), and the International Union of the Microbial Societies recently issued recommendations to preserve microbial diversity (16), to name a few collective efforts for urgent reform. Microbiome stewardship expands these ideas to encompass globally interconnected microbial habitats and their collective contributions to planetary health. Importantly, a clearly articulated definition and guiding principles of microbiome stewardship would provide a foundation for policy action across diverse regulatory domains, such as public health, agriculture, and the environment.

By stewardship, we refer to a sense of duty to preserve and care for something that holds intrinsic and utilitarian value. Microbiome stewardship involves the recognition of microbiomes as an essential foundation for life on Earth and entails conscious and strategic conservation practices to limit the deterioration of health in humans, nonhuman life, and the environment. Effective microbiome stewardship requires collective societal action at all levels: institutional practices (e.g., guidelines for prophylactic antibiotic use), national policy (e.g., regulation of agricultural pesticides known to harm microbial ecologies), and global standards (e.g., international agreements on microplastics). Implementing microbiome stewardship at any of these levels is challenging but absolutely essential to safeguard the well-being of future generations.

Microbiomes are critical for our existence, and an important goal of stewardship will be to preserve key elements that are seen as beneficial (17). In contrast to other valuable natural resources (e.g., air, water, and environments), it is difficult to define what, precisely, should be preserved in the context of microbiomes. Microbiomes are dynamic and largely unpredictable systems comprised of bacteria, fungi, archaea, protozoa, and viruses—many of which remain woefully under-characterized. Microbiomes are inherently functionally redundant, yet not all communities of similar microbiota or microbial exposures yield the same functional outcomes in all hosts (including humans) or environments. If microbiomes are a communal environmental resource (17), how do we define what microbes, functions, and ecological interactions are worth preserving? How do we assign intrinsic value to rare microbiota? How can we move beyond simplistic notions of what is a "good" versus "bad" microbiome and towards developing approaches to quantify these concepts in a meaningful way?

It is critical that collective microbiomes are protected through policy at the societal level, and these efforts recognize that existing social inequities may be exacerbated through diverse microbial exposures. We therefore need a unifying concept emphasizing the importance of microbiomes for human and environmental health that can be applied across policy domains. Motivated by these questions, the concept of microbiome stewardship recognizes the necessity of microbial communities in sustaining human and ecosystem health and emphasizes the imperative to protect them (17).

## CHALLENGES TO DEVELOPING MICROBIOME STEWARDSHIP

Although the notion of microbiome stewardship has intuitive appeal, it currently lacks precision, as several scholars in recent years have referred to it with notable differences. O'Doherty et al. discussed microbiome stewardship in the context of public health, noting that all humans share a microbial commons that needs to be guarded against mismanagement (17). Choudoir and Eggleston, in contrast, focused mainly on preserving environmental microbiomes, highlighting the connections with environmental justice (4). Notably, both O'Doherty et al. and Choudoir and Eggleston point to the importance of public and stakeholder engagement in microbiome stewardship. Taking a different approach, Peixoto et al. (18) propose a "careful and responsible management of ecosystem resources using the microbiome (termed microbiome stewardship) to rehabilitate organisms and ecosystem functions." Peixoto et al.'s conception of microbiome stewardship is thus very different. It does not focus on the preservation of microbiota but rather the use of microbial interventions to steward other plant and wildlife species (18).

The full realization of the value of microbiome stewardship will rely on a generally agreed-upon definition to include in policy documents and mission statements of organizations tasked with protecting human, nonhuman, and environmental health (19). The first challenge in developing effective microbiome stewardship is therefore to develop a meaningful and robust definition of the concept, with broad and iterative input from a range of experts and stakeholders.

A second challenge in developing microbiome stewardship is the difficulty of specifying what microbial taxa or microbial function is to be stewarded within a given context. We may consider the "health" of a microbiome in terms of its ability to carry

out crucial ecosystem functions, such as short-chain fatty acid production in the gut or nitrogen fixation in soil. Adopting a functional lens is no easy task as most microbiomes are complex, with potentially thousands of species carrying out cryptic functions across spatial and temporal scales. Due to the emergent properties of microbiomes, it is difficult to predict which microbiota may achieve a desired outcome in a given ecosystem or circumstance. This necessitates a comprehensive survey of current methodologies in defining and comparing the diversity of microbiome samples and how these are mapped to perceived "healthy" and "unhealthy" states (i.e., how can, are, and should they be assessed?). This also requires an examination of microbial connectivity between habitats, patterns of dispersal, and community resilience following environmental disturbances. Effective microbiome stewardship principles must be founded in microbial ecology, evolution, and biogeography.

A third challenge is to ensure ethical practice while respecting resource and data sovereignty. Arguments to sample and hoard microbial diversity for future use (i.e., biobanking) have grown in part from observations of biodiversity losses due to homogenization in the gut by ultra-processed diets or in soils through intensive land use and environmental degradation. This has led some researchers to seek out the microbiomes of Indigenous and remote communities to develop technologies for replenishing or restoring the microbiomes of affluent populations. It is critical that efforts to preserve or restore the microbiomes of some populations do not involve the exploitation of Indigenous and other communities and that data curation of environmental landscapes and ecological knowledge follows ethical practices (20).

## INTEGRATING MICROBIOME STEWARDSHIP WITH EXISTING REGULATORY APPROACHES

Implementing microbiome stewardship requires significant shifts in policy frameworks, as it would require setting benchmarks for "healthy'" microbiome profiles and monitoring select pathogenic or deleterious microbes. We expect some of the biggest changes in public health policies. Microbiomes are implicated in an increasing number of chronic illnesses, many of which fall under the public health umbrella (e.g., certain cancers and metabolic and immune disorders). One could envisage public health programs such as food fortification for healthy microbiomes, consideration of microbiomes in built environments (e.g., NSF Engineering Research Center PreMiEr, Durham, NC), and attention to green spaces in cities for purposes of healthy microbiome exposure.

Another important policy domain for centering microbiomes is agriculture and food production, namely the use of biocidal chemicals (i.e., pesticides, herbicides, and fungicides). While opinions on optimal regulatory oversight differ significantly, it is safe to say that the conservation of soil microbes beneficial for human health is currently not high on the policy agenda. There is a similar neglect of microbial ecologies of insects, plants, and other organisms that contribute to diverse food systems (3).

## CONCLUSION

We intend to inspire and motivate broader input in working toward microbiome stewardship, particularly in enhancing its practical value. The first step will be to develop a framework, a broadly acceptable definition, and guiding principles of microbiome stewardship. A continued dialogue, backed by case study data, will promote the idea of microbiome stewardship among key leaders most likely to establish its implementation and quantitative assessment. This cannot be based only on the actions of individuals; stewardship requires a reimagining of human and environmental health as a collective phenomenon and microbiome protection across institutional and policy levels. In this short article, we have focused mainly on the role of microbiomes in human health. Ultimately, however, we hold that the Earth's microbiomes and microbial ecologies are worth preserving not only for their utility for humans. We believe microbiome stewardship is vital for sustaining all life on Earth, and achieving this practice will be a huge undertaking. We have articulated the need for microbiome stewardship

as a foundational concept across multiple policy domains while also identifying key challenges that will need to be overcome to meaningfully develop and implement principles of microbiome stewardship.

## ACKNOWLEDGMENTS

We thank Lola Holcomb for their helpful feedback and organizational contributions to this manuscript.

This work was supported by the United States Department of Agriculture National Institute of Food and Agriculture Hatch Project Accession 7004439 (M.J.C.), the United States Department of Agriculture National Institute of Food and Agriculture through the Maine Agricultural & Forest Experiment Station: Hatch Project ME022329 (S.L.I.), the National Institute of Health (NIH/NIDDK 1R15DK133826-01) (S.L.I.), the Canadian Natural Sciences and Engineering Research Council (R.G.B.), Canada Research Chairs program (E.A.-V.), the Canadian Institutes of Health Research (Funding Reference Number: 191753) (K.C.O.), and the University of Guelph Institute for Environmental Research (K.C.O.).

Conceptualization: M.J.C., S.L.I., R.G.B., D.S.S., E.A.-V., K.C.O. Visualization: M.J.C. Funding acquisition: K.C.O. Project administration: K.C.O., S.L.I., R.G.B. Writing–original draft: M.J.C., K.C.O. Writing–review and editing: M.J.C., S.L.I., R.G.B., D.S.S., E.A.-V., K.C.O.

## AUTHOR AFFILIATIONS

[1]Department of Plant and Microbial Biology, North Carolina State University, Raleigh, North Carolina, USA

[2]School of Food and Agriculture, University of Maine, Orono, Maine, USA

[3]Faculty of Computer Science and Institute for Comparative Genomics, Dalhousie University, Halifax, Nova Scotia, Canada

[4]Faculty of Medicine and Health, University of Sydney School of Public Health, Camperdown, New South Wales, Australia

[5]Department of Molecular and Cellular Biology, University of Guelph, Guelph, Ontario, Canada

[6]Department of Psychology, University of Guelph, Guelph, Ontario, Canada

## AUTHOR ORCIDs

Mallory J. Choudoir  http://orcid.org/0000-0002-9117-5150
Suzanne L. Ishaq  http://orcid.org/0000-0002-2615-8055
Robert G. Beiko  http://orcid.org/0000-0002-5065-4980
Emma Allen-Vercoe  http://orcid.org/0000-0002-8716-327X
Kieran C. O'Doherty  http://orcid.org/0000-0002-9242-2061

## FUNDING

| Funder | Grant(s) | Author(s) |
| --- | --- | --- |
| Canadian Institutes of Health Research | 191753 | Kieran C. O'Doherty |
| U.S. Department of Agriculture's National Institute of Food and Agriculture | 7004439, ME022329 | Mallory J. Choudoir |
| National Institutes of Health | NIH/NIDDK 1R15DK133826-01 | Suzanne L. Ishaq |
| Natural Sciences and Engineering Research Council of Canada | | Robert G. Beiko |
| Canada Research Chairs | | Emma Allen-Vercoe |
| University of Guelph Institute of Environmental Research | | Kieran C. O'Doherty |

## ADDITIONAL FILES

The following material is available online.

### Open Peer Review

**PEER REVIEW HISTORY (review-history.pdf).** An accounting of the reviewer comments and feedback.

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
