## [Reviewer comments · mSystems]

The case for microbiome stewardship: What it is and how to get there

Mallory Choudoir, Suzanne Ishaq, Robert Beiko, Diego Silva, Emma Allen-Vercoe, and Kieran O'Doherty

Corresponding Author(s): Kieran O'Doherty, University of Guelph

Review Timeline:

Submission Date:	January 13, 2025
Editorial Decision:	February 13, 2025
Revision Received:	March 7, 2025
Accepted:	March 10, 2025

Editor: Jack Gilbert

Reviewer(s): Disclosure of reviewer identity is with reference to reviewer comments included in decision letter(s). The following individuals involved in review of your submission have agreed to reveal their identity: Lita Proctor (Reviewer #1)

Transaction Report:

DOI: <https://doi.org/10.1128/msystems.00062-25>

Re: mSystems00062-25 (The case for microbiome stewardship: What it is and how to get there)

Dear Prof. Kieran C O'Doherty:

Revision Guidelines

- Upload point-by-point responses to the issues raised by the reviewers in a file named "Response to Reviewers," NOT in your cover letter.
- Upload a compare copy of the manuscript (without figures) as a "Marked-Up Manuscript" file.
- Upload a clean .DOC/.DOCX version of the revised manuscript and remove the previous version.
- Each figure must be uploaded as a separate, editable, high-resolution file (TIFF or EPS preferred), and any multipanel figures must be assembled into one file.

Minireviews are not subject to publication charges.

Author Bios: We encourage you to submit a biographical sketch of each author (limit of 150 words) along with a photo to be published at the end of your article. You can submit these with your modified manuscript.

Figures Enhancement: ASM has engaged a professional science illustrator, Patrick Lane of ScEYence Studios, to work with minireview authors at the modification stage to generate improved figures that are uniform throughout the journal. This art enhancement service is free of charge to authors of minireviews and full-length reviews, and turnaround time is fast. I think you will be pleased with the results. Please contact Patrick on receiving this letter. Complete contact information for Patrick and further instructions are posted at <https://journals.asm.org/pb-assets/pdf-text-excel-files/graphical-enhancement-support.pdf>.

Sincerely,
Jack Gilbert
Editor
mSystems

Reviewer #1 (Comments for the Author):

These authors are a part of a growing call from the microbial sciences communities for promoting the roles of and protection of microbiomes of all systems for the health of all on this planet. They rightly identify the difficulty in defining what is being protected as microbiomes are far more complex than the sum of their members. What I think is lacking here is an acknowledgement that, though they sound alone in these efforts, in fact they are not alone in championing the stewardship of the microbiome. I would ask the co-authors to consider adding some text highlighting some of these other calls and to include one or more of the references below. I also think that since these co-authors represent many countries, their efforts should also include many countries as microbes know no borders. (By the way, I am sharing these references because a group of us have a ms under consideration to announce the launch of an int'l forum "World Microbiome Partnership" which is advocating 'the power of the microbe' for resolving many of the issues this stewardship group is calling out. But happy to share these refs if of use to these authors)

A few other remarks:

- a. Please add the IPCC which only recently recognized the role of microbes in climate change. In fact, most of the int'l initiatives they list only recently recognized the role of microbes in their efforts (which I think has been part of the problem - a lack of understanding on the foundational role of microbes in all systems, which is on the microbiological community to rectify)
- b. For microbiome ecosystem functions, please add 'base of the food web' or 'food source' and also production of critical metabolites/molecules (like SCFAs or neurotransmitters in guts or nitrogen in soils/oceans) to their list.
- c. Food production practices should include some specifics like addition of emulsifiers and other additives which either alter gut microbiome functions or bypass the normal digestion system altogether
- d. Please also add specifics on food production like loss of soil microbial diversity leads to pathogen exposure in monoculture crops and thereby require agrochemicals like pesticides/herbicides

Some suggested references to include:

1. Anand, S. et al. Weaponizing microbes for peace. *Microbial Biotech.* 16, 1091-1111 (2023).
2. D'Hondt, K. et al. Microbiome innovations for a sustainable future. *Nature Microbiol.* 6, 138-142 (2021).
3. Toju, H. et al. Core microbiomes for sustainable agroecosystems. *Nature Plants* 4, 247-257 (2018).
4. Pexioto, R. A. et al. Harnessing the microbiome to prevent global biodiversity loss. *Nature Microbiol.* 7, 1726-1735 (2022).
5. Cavicchioli, R. et al. Scientists' warning to humanity: microorganisms and climate change. *Nature Reviews Microbiol.* 17, 569-585 (2019).
6. Tiedje, J. Microbes and climate change: a research prospectus for the future. *mBio* 13, 1-9 (2022).
7. Crowther, T.W. et al. Scientists' call to action: Microbes, planetary health, and the Sustainable Development Goals. *Cell* 187, 5195 - 5216 (2024).
8. Rappuoli, R. et al. Save the microbes to save the planet. A call to action of the International Union of Microbiological Societies (IUMS). *One Health Outlook* 5, 1-5 (2023).

Reviewer #2 (Comments for the Author):

This manuscript presents an important and timely discussion, proposing the concept of "microbiome stewardship" as a unifying idea that 1. can emphasize the importance of microbiomes for human and environmental health and 2. can act across different domains (agriculture, food systems, health etc) to advocate for policy interventions in favour of healthy microbiomes. The problem the paper addresses is the lack of policy initiatives focused on microbial ecologies.

The work is divided into three main parts: 1. An overview of evidence suggesting the deterioration of microbiomes across systems and the consequent effects on human and environmental health; 2. Outline of the current policy gaps; 3. Examples of possible policy interventions within existing regulatory approaches.

This manuscript offers a timely discussion on the role of microbiomes in human, non-human and environmental health. I recommend strengthening the argument by addressing at least comments 1 and 2 (see below) before publication.

Strengths:

The novelty of the article is in the suggestion of the general framework of stewardship, which they define as "a sense of duty to preserve and care for something that holds intrinsic and utilitarian value" (line 104-5) to address a complex issue that pertains to multiple domains. I also really appreciate the shift of focus from the individual level (individual choice, or medical interventions on patients with specific conditions) to a more systemic level that recognizes how individual choices and behaviours are always enabled and at the same time constrained by a broader economic, geopolitical, historical and cultural context - it is precisely at this systemic level that the authors situate their argument.

Areas in which the manuscript could be improved:

1. While the manuscript introduces "microbiome stewardship" as a key concept, it lacks a precise and operational definition. The authors acknowledge that this remains a challenge to be addressed in the future, but I believe at least a tentative and provisional definition is necessary to make a strong case with policymakers and researchers alike.

(The concept of stewardship has a long history in environmentalist movements, I'm thinking in particular about the environmental stewardship as promoted by Aldo Leopold, or the counterculture's idea of planetary stewardship as advocated for example by Buckminster Fuller in the *Manual for Spaceship Earth*. In this latter context, environmentalist goals were also mixed with a good dose of technological optimism. I wonder to what extent their use of the term stewardship recalls these connotations or departs from them. A particular link that I am thinking of might be OneHealth initiatives, in which planetary health is tied to the health of its microbiome. These approaches adopt a rather "global" perspective, which might not necessarily fit well with a policy-oriented approach that recognizes local specificities. Better positioning this intervention with respect to these previous uses of "stewardship" in ecological context can be a possible starting point to clarify the definition and scope of microbial stewardship)

2. The concept of an "healthy microbiome" is foundational for the argument and, more in general, its characterization is crucial for the implementation of policy interventions. The authors well outline the challenges that are related to its definition (line 113-

124 and 141-148), and yet they do not advance any suggestions on how the present uncertainties related to the concept might be addressed. I think a preliminary set of recommendations would suffice to lay the foundations on which further conversations can be built.

I'd also like to offer a couple of minor suggestions on how the paper might be strengthened:

3. The paper seems to well support the idea that microbiomes have a utilitarian value ("microbiomes are critical for our existence" line 113), and yet, the authors also mention an intrinsic value (lines 105 and 185) which seems - in this work - ideological or at least not adequately substantiated.

4. The section titled "integrating microbiome stewardship with existing regulatory approaches" could be expanded with examples of similar interventions that can work as models (if any relevant example exists, obviously) and/or with the discussion of potential solutions (for example specific policy instruments).

We enthusiastically thank the reviewers for supporting this discussion and for the helpful feedback. We have addressed all reviewer comments directly below, and have integrated revisions into the current version of the manuscript. We have also made minor revisions throughout to correct small typos and improve clarity (see _MARKEDUPMANUSCRIPT document for all revisions).

Reviewer #1 (Comments for the Author):

These authors are a part of a growing call from the microbial sciences communities for promoting the roles of and protection of microbiomes of all systems for the health of all on this planet. They rightly identify the difficulty in defining what is being protected as microbiomes are far more complex than the sum of their members. What I think is lacking here is an acknowledgement that, though they sound alone in these efforts, in fact they are not alone in championing the stewardship of the microbiome. I would ask the co-authors to consider adding some text highlighting some of these other calls and to include one or more of the references below. I also think that since these co-authors represent many countries, their efforts should also include many countries as microbes know no borders. (By the way, I am sharing these references because a group of us have a ms under consideration to announce the launch of an int'l forum "World Microbiome Partnership" which is advocating 'the power of the microbe' for resolving many of the issues this stewardship group is calling out. But happy to share these refs if of use to these authors)

We agree with the reviewer – that we are not the lone champion of microbiome protection and conservation. We have expanded the discussion to explicitly include this acknowledgement, and have also included some of the citations suggested (i.e., Cavicchiolo et al. 2019 and Rappuoli et al. 2023). However, we believe that our concept of microbiome stewardship is unique in that we aim to propose a unified framework of assessment to be integrated across policy domains. We have added the following text:

“We are not lone champions of microbial protection and conservation. “One Health” principles recognise the importance of microbes in human, animal, and plant health. Microbiologists and climate change experts have issued a Consensus Statement calling attention to the role of microbes in climate change responses (15), and the International Union of the Microbial Societies recently issued recommendations to preserve microbial diversity (16), to name a few collective efforts for urgent reform. Microbiome stewardship expands these ideas to encompass globally interconnected microbial habitats and their collective contributions to planetary health. Importantly, a clearly articulated definition and guiding principles of microbiome stewardship would provide a foundation for policy action across diverse regulatory domains, such as public health, agriculture, and the environment.”

A few other remarks:

a. Please add the IPCC which only recently recognized the role of microbes in climate change. In fact, most of the int'l initiatives they list only recently recognized the role of microbes in their efforts (which I think has been part of the problem - a lack of understanding on the foundational role of microbes in all systems, which is on the microbiological community to rectify).

We've revised this list to include IPCC. Thank you, that was an important oversight on our part.

“Although the importance of microbes is recently recognised by several high-level agreements and initiatives (e.g., United Nations Convention of Biological Diversity, Food and Agricultural Organization International Network on Soil Biodiversity, Intergovernmental Panel on Climate Change, One Health approaches), we lack a comprehensive understanding of the explicit roles microbes occupy across ecological and temporal scales.”

b. For microbiome ecosystem functions, please add 'base of the food web' or 'food source' and also production of critical metabolites/molecules (like SCFAs or neurotransmitters in guts or nitrogen in soils/oceans) to their list.

We've expanded our list to include these important functions.

“The central importance of microbiomes and their ecological functions (e.g, global nutrient cycling, decomposition, bioremediation, pathogen reduction, and as a source of food/nutrition and critical metabolites/molecules) for the health and well-being of ecosystem, animal, plant, and human health is now well established.”

c. Food production practices should include some specifics like addition of emulsifiers and other additives which either alter gut microbiome functions or bypass the normal digestion system altogether.

We've added an additional example in this section about dietary emulsifiers and their links to inflammation/metabolic syndrome.

“In food and nutrition policy, attention is focused on macronutrients, yet evidence suggests that food production practices shape the microbiomes of foods and human microbiomes with no awareness of these dynamics in food or agricultural policies. Growing evidence shows that chronic exposure to agricultural pesticides (i.e., insecticides, herbicides, and fungicides) leads to gut microbiome disturbances (13). Food additives such as emulsifiers cause changes in gut microbiomes that are linked to inflammation and metabolic syndrome (14). Conversely, for example, traditionally (Indigenous Greenlandic) prepared foods have much richer microbiome profiles than the same foods prepared with industrial methods (2).”

d. Please also add specifics on food production like loss of soil microbial diversity leads to pathogen exposure in monoculture crops and thereby require agrochemicals like pesticides/herbicides

We've also expanded discussion in this section to highlight how pesticide use leads to microbiome disturbances. See text inclusion above.

Some suggested references to include:

1. Anand, S. et al. Weaponizing microbes for peace. *Microbial Biotech.* 16, 1091-1111 (2023).
2. D'Hondt, K. et al. Microbiome innovations for a sustainable future. *Nature Microbiol.* 6, 138-142 (2021).
3. Toju, H. et al. Core microbiomes for sustainable agroecosystems. *Nature Plants* 4, 247-257 (2018).
4. Pexioto, R. A. et al. Harnessing the microbiome to prevent global biodiversity loss. *Nature Microbiol.* 7, 1726-1735 (2022).

5. Cavicchioli, R. et al. Scientists' warning to humanity: microorganisms and climate change. *Nature Reviews Microbiol.* 17, 569-585 (2019).
6. Tiedje, J. Microbes and climate change: a research prospectus for the future. *mBio* 13, 1-9 (2022).
7. Crowther, T.W. et al. Scientists' call to action: Microbes, planetary health, and the Sustainable Development Goals. *Cell* 187, 5195 - 5216 (2024).
8. Rappuoli, R. et al. Save the microbes to save the planet. A call to action of the International Union of Microbiological Societies (IUMS). *One Health Outlook* 5, 1-5 (2023).

Reviewer #2 (Comments for the Author):

This manuscript presents an important and timely discussion, proposing the concept of "microbiome stewardship" as a unifying idea that 1. can emphasize the importance of microbiomes for human and environmental health and 2. can act across different domains (agriculture, food systems, health etc) to advocate for policy interventions in favour of healthy microbiomes. The problem the paper addresses is the lack of policy initiatives focused on microbial ecologies.

The work is divided into three main parts: 1. An overview of evidence suggesting the deterioration of microbiomes across systems and the consequent effects on human and environmental health; 2. Outline of the current policy gaps; 3. Examples of possible policy interventions within existing regulatory approaches. This manuscript offers a timely discussion on the role of microbiomes in human, non-human and environmental health. I recommend strengthening the argument by addressing at least comments 1 and 2 (see below) before publication.

Strengths:

The novelty of the article is in the suggestion of the general framework of stewardship, which they define as "a sense of duty to preserve and care for something that holds intrinsic and utilitarian value" (line 104-5) to address a complex issue that pertains to multiple domains. I also really appreciate the shift of focus from the individual level (individual choice, or medical interventions on patients with specific conditions) to a more systemic level that recognizes how individual choices and behaviours are always enabled and at the same time constrained by a broader economic, geopolitical, historical and cultural context - it is precisely at this systemic level that the authors situate their argument.

Areas in which the manuscript could be improved:

1. While the manuscript introduces "microbiome stewardship" as a key concept, it lacks a precise and operational definition. The authors acknowledge that this remains a challenge to be addressed in the future, but I believe at least a tentative and provisional definition is necessary to make a strong case with policymakers and researchers alike. (The concept of stewardship has a long history in environmentalist movements, I'm thinking in particular about the environmental stewardship as promoted by Aldo Leopold, or the counterculture's idea of

planetary stewardship as advocated for example by Buckminster Fuller in the Manual for Spaceship Earth. In this latter context, environmentalist goals were also mixed with a good dose of technological optimism. I wonder to what extent their use of the term stewardship recalls these connotations or departs from them. A particular link that I am thinking of might be OneHealth initiatives, in which planetary health is tied to the health of its microbiome. These approaches adopt a rather "global" perspective, which might not necessarily fit well with a policy-oriented approach that recognizes local specificities. Better positioning this intervention with respect to these previous uses of "stewardship" in ecological context can be a possible starting point to clarify the definition and scope of microbial stewardship)

Yes indeed! Very briefly, one of our initial tasks for the Microbiome Stewardship project is to consider concepts of conservation and stewardship across diverse fields, movements, and philosophies. The One Health concept aligns well with microbiome stewardship (we have included One Health Initiatives in our list of groups that recognize the importance of microbes), as the reviewer suggests, our efforts aim to target policies across domains and scales.

While a comprehensive review of stewardship and conservation definitions is outside of the scope of this perspective, we did expand the discussion of "microbiome stewardship" definitions in the current literature.

"Although the notion of microbiome stewardship has intuitive appeal, it currently lacks precision as in recent years several scholars have referred to it with notable differences. O'Doherty *et al.* discussed microbiome stewardship in the context of public health, noting that all humans share a microbial commons that needs to be guarded against mismanagement (17). Choudoir & Eggleston, in contrast, focused mainly on preserving environmental microbiomes, highlighting connections with environmental justice (4). Notably, both O'Doherty *et al.* and Choudoir & Eggleston point to the importance of public and stakeholder engagement in microbiome stewardship. Taking a different approach, Peixoto *et al.* (18) propose "careful and responsible management of ecosystem resources using the microbiome (termed microbiome stewardship) to rehabilitate organisms and ecosystem functions." Peixoto *et al.*'s conception of microbiome stewardship is thus very different; it does not focus on the preservation of microbiota, but rather the use of microbial interventions to steward other plant and wildlife species (18)."

2. The concept of an "healthy microbiome" is foundational for the argument and, more in general, its characterization is crucial for the implementation of policy interventions. The authors well outline the challenges that are related to its definition (line113-124 and 141-148), and yet they do not advance any suggestions on how the present uncertainties related to the concept might be addressed. I think a preliminary set of recommendations would suffice to lay the foundations on which further conversations can be built.

We have added a short section emphasizing the importance of metrics for evaluating "healthy" and "unhealthy" states must be rooted in ecological understanding of microbiome dynamics.

"This necessitates a comprehensive survey of current methodologies in defining and comparing the diversity of microbiome samples and how these are mapped to perceived "healthy" and "unhealthy" states (i.e., how **can-**, **are-**, and **should** they be assessed?). This also requires examination of microbial connectivity between habitats, patterns of dispersal, and community resilience following environmental

disturbances. Effective microbiome stewardship principles must be founded in microbial ecology, evolution, and biogeography.”

I'd also like to offer a couple of minor suggestions on how the paper might be strengthened:

3. The paper seems to well support the idea that microbiomes have a utilitarian value ("microbiomes are critical for our existence" line 113), and yet, the authors also mention an intrinsic value (lines 105 and 185) which seems - in this work - ideological or at least not adequately substantiated.

We agree with the reviewer's assessment that we have not yet developed a strong argument for the intrinsic value of microbes, as we felt that that information exists in literature already. Here, we focus on microbes with clear connections to human or environmental health, although we suggest the possibility that all microbiota and their ecologies contribute to planetary health. The question being, do we prioritize human health above planetary health? This is a concept we are excited to explore in subsequent manuscripts.

4. The section titled "integrating microbiome stewardship with existing regulatory approaches" could be expanded with examples of similar interventions that can work as models (if any relevant example exists, obviously) and/or with the discussion of potential solutions (for example specific policy instruments).

We are not aware of regulatory approaches that include microbiome stewardship practice and demonstrate beneficial health outcomes. Indeed, this is in large part the impetus for the work we are advocating for in the paper. Due to space and scope limitations, we have not added to this section, though we hope to have a lot more to say about this over the next few years.

Re: mSystems00062-25R1 (The case for microbiome stewardship: What it is and how to get there)

Dear Prof. O'Doherty:

Your manuscript has been accepted, and I am forwarding it to the ASM production staff for publication. Your paper will first be checked to make sure all elements meet the technical requirements. ASM staff will contact you if anything needs to be revised before copyediting and production can begin. Otherwise, you will be notified when your proofs are ready to be viewed.

Sincerely,
Jack Gilbert
Editor